# The Microenvironment of Pituitary Tumors—Biological and Therapeutic Implications

**DOI:** 10.3390/cancers11101605

**Published:** 2019-10-21

**Authors:** Mirela Diana Ilie, Alexandre Vasiljevic, Gérald Raverot, Philippe Bertolino

**Affiliations:** 1Cancer Research Centre of Lyon (CRCL), INSERM U1052, CNRS UMR5286, Claude Bernard University, 69008 Lyon, France, mireladiana.ilie@gmail.com (M.D.I.); alexandre.vasiljevic@chu-lyon.fr (A.V.); 2“Claude Bernard” Lyon 1 University, University of Lyon, 69100 Villeurbanne, France; 3Endocrinology Department, “C.I.Parhon” National Institute of Endocrinology, 011863 Bucharest, Romania; 4Pathology Department, “Groupement Hospitalier Est” Hospices Civils de Lyon, 69677 Bron, France; 5Endocrinology Department, “Groupement Hospitalier Est” Hospices Civils de Lyon, 69677 Bron, France

**Keywords:** pituitary adenoma, pituitary tumor, pituitary carcinoma, tumor microenvironment, angiogenesis, immune infiltrate, folliculostellate cells, extracellular matrix, treatment, immunotherapy

## Abstract

The tumor microenvironment (TME) includes resident and infiltrative non-tumor cells, as well as blood and lymph vessels, extracellular matrix molecules, and numerous soluble factors, such as cytokines and chemokines. While the TME is now considered to be a prognostic tool and a therapeutic target for many cancers, little is known about its composition in pituitary tumors. This review summarizes our current knowledge of the TME within pituitary tumors and the strong interest in TME as a therapeutic target. While we cover the importance of angiogenesis and immune infiltrating cells, we also address the role of the elusive folliculostellate cells, the emerging literature on pituitary tumor-associated fibroblasts, and the contribution of extracellular matrix components in these tumors. The cases of human pituitary tumors treated with TME-targeting therapies are reviewed and emerging concepts of vascular normalization and combined therapies are presented. Together, this snapshot overview of the current literature pinpoints not only the underestimated role of TME components in pituitary tumor biology, but also the major promise it may offer for both prognosis and targeted therapeutics.

## 1. Introduction

Pituitary tumors are common tumors of the anterior pituitary gland that can appear in up to a fifth of the general population [1,2]. Based on the sole clinically relevant anterior pituitary tumors, their prevalence is 80–100 cases/100,000 people. Pituitary carcinomas are rare and represent 0.2% of pituitary tumors [3]. While most cases of pituitary tumor present as slowly progressing tumors that can be easily treated by surgery or drugs, some present as aggressive tumors whose behavior is difficult to predict [3,4,5]. Clinical management of aggressive pituitary tumors is moreover difficult due to their resistance and/or multiple recurrences despite the correct use of surgery, conventional medical treatment, and radiotherapy [3]. Therefore, the identification of factors leading to an aggressive phenotype and the discovery of new therapeutic targets/treatment options are of particular importance for the understanding of the biology of pituitary tumors and the management of aggressive pituitary tumors, including carcinomas.

It is now accepted that, within tumors, transformed cells can recruit and corrupt other cell-types and that the interactions between tumor cells and non-tumor cells create and shape the tumor microenvironment (TME). The TME is the environment within which the tumor exists, and includes structures such as the blood and lymph vessels; resident and infiltrating non-tumor cells like tumor-associated fibroblasts and immune cells; and non-cellular components including numerous molecules, such as cytokines and extracellular matrix-components. While the existence and importance of the TME as both a prognostic tool and a therapeutic target is well-accepted and has been described regarding numerous cancers [6,7], research on the pituitary tumor TME is still scarce. Here, we provide an overview of the current known composition and function of the microenvironment of pituitary tumors and of its importance as a therapeutic target.

## 2. Angiogenesis in the Tumor Microenvironment

### 2.1. The Implications of Angiogenesis in Pituitary Tumors

The pivotal role of angiogenesis (the formation of new blood vessels from pre-existing ones [8]) on tumorigenesis-related processes was already starting to be recognized in 1971 when Folkman proposed the use of anti-angiogenic agents for targeting the interactions between tumor cells and the TME [9,10]. While it is accepted that angiogenesis is a highly complex process implicating numerous molecules and signaling pathways, the vascular endothelial growth factor (VEGF)/VEGF-receptor (VEGFR) pathway is central to angiogenesis and remains the most studied and targeted so far [8,11]. When examining VEGF expression using immunohistochemistry (IHC) in a series of 148 pituitary adenomas (PAs) and six pituitary carcinomas, Lloyd et al. found a decreased expression in PAs compared to the adjacent normal pituitary tissue and an increased expression in pituitary carcinomas compared to PAs, suggesting an up-regulation and a potential role for VEGF in tumor progression and malignant transformation [12]. These findings are consistent with the observations of other authors, who reported reduced vascular densities in PAs, using Factor VIII immunoreactivity as a marker, compared to normal pituitary tissue in a large cohort of samples (103 PAs, 8 primary pituitary carcinomas, and 20 normal pituitary glands) [13]. These authors also noticed that in pituitary carcinomas, the vascular density was higher than in PAs when only “hot spots” (i.e., areas of focally increased vascularization) were considered, but was similar to that of PAs when the whole lesion was taken into account [13]. Vidal et al., who assessed microvessel densities by performing IHC for the CD34 antigen (as a marker of endothelial cells) in a series of 157 PAs and 7 pituitary carcinomas, also found the highest percentage of microvessel density was in pituitary carcinomas [14]. While these observations support the view that PAs are generally less vascularized than the normal pituitary gland and that pituitary carcinomas are more vascularized than PAs, the comparative analysis of tumor angiogenesis between different PA subtypes has yielded contradictory results, as different studies have reported divergent findings regarding which hormonal subtype PA appears to be more vascularized, and also regarding the association between angiogenesis and clinical traits, such as invasiveness [12,13,14,15,16].

### 2.2. Therapeutic Targeting of Angiogenesis in Pituitary Tumors

In terms of the therapeutic targeting of angiogenesis, the first approved anti-angiogenic drug, bevacizumab, a monoclonal antibody directed against VEGF [17], has thus far been the drug most used for targeting the VEGF pathway in pituitary tumors. The published clinical cases have been recently reviewed by our team [18] and are presented in Table 1 [3,19,20,21,22,23,24,25]. Twelve cases treated with bevacizumab have so far been reported, either as a monotherapy or in different combinations, with either pasireotide or temozolomide (TMZ)—alone or with the Stupp protocol (i.e., the administration of fractionated radiotherapy concomitantly with daily TMZ 75 mg/m^2^, followed by 6–12 months of adjuvant TMZ 150–200 mg/m^2^ for five days every 28 days). Bevacizumab was administered in 10 patients who previously received TMZ, and in 2 patients, it was associated with first-line TMZ. Of the 12 cases, 5 were corticotroph tumors, including 4 carcinomas and 1 adenoma, and 7 were undetermined pituitary tumors. Interestingly, a partial remission was observed in 3 out of 12 cases, while a complete remission, with no recurrence for 5 years, was seen in a case of corticotroph carcinoma where bevacizumab was associated with first-line TMZ with a Stupp protocol. Moreover, stable disease was seen in 5 cases (4 corticotroph tumors and 1 unknown subtype), whereas progressive disease was observed in only 2 cases [18]. In parallel with bevacizumab, the use of the sunitinib and axitinib tyrosine kinase inhibitors have been proposed as alternative therapeutic options that target the VEGF/VEGFR pathway [26]. Tyrosine kinases are a family of enzymes, to which the VEGFRs belong [8,27], that catalyze the phosphorylation of many important proteins that are implicated in a variety of physiological and pathological processes, including cell proliferation, apoptosis, tumor invasion, metastasis, and angiogenesis [28,29]. Therefore, the anti-tumoral effects of tyrosine kinase inhibitors are achieved through a number of mechanisms including the inhibition of tumor cell repair, induction of apoptosis, and via anti-angiogenetic effects [26]. To date, the use of sunitinib has only been reported in one case of an undetermined pituitary tumor previously treated with TMZ, and the patient showed progressive disease on sunitinib [23].

Although these cases are too limited in number to draw any conclusion about the effect of targeting the VEGF/VEGFR pathway in pituitary tumors, the experience from other cancers showed that tumors may be inherently resistant or become resistant to VEGF/VEGFR inhibition, and moreover, that targeting the VEGF/VEGFR pathway may increase intra-tumor hypoxia in a dose- and time-dependent manner [8]. In fact, the pathological angiogenesis associated with tumors, resulting from an imbalance between pro-angiogenic and anti-angiogenic factors, gives rise to abnormal vessels. The immaturity, disorganization, and increased permeability of these abnormal vessels leads to high pressure in the interstitial fluid, poor tumor perfusion, and increased hypoxia, which in turn can facilitate the selection of aggressive tumor cells, reduce the anti-tumoral immune response, impede the penetration and the efficacy of chemotherapeutic drugs, and lower the efficacy of radiotherapy. The concept of vascular normalization has emerged from these observations, and represents a therapeutic strategy aiming at restoring proper perfusion and oxygenation of the tumor tissue, in order to limit the selection of aggressive tumor cells, to facilitate the anti-tumoral immune response, and improve the action of chemotherapeutic drugs [8,30].

Strategies to achieve vascular normalization have been reviewed by Viallard and Larrivée [30], and include the careful adaptation of the dose and the time frame for the administration of drugs targeting the VEGF/VEGFR pathway [8,30]. Another proposed option for the normalization of vascular structure and function is to use dopamine receptor type 2 agonists, which have been shown to act on both endothelial cells and pericytes [30,31,32]. The effect of dopamine receptor type 2 agonists on blood vessel normalization was recently tested in a mouse model of hemorrhagic prolactinoma that presented an aberrant arterial vascularization with altered blood vessels, leading to a loss of dopamine supply [33]. In this model, administration of bromocriptine alone or in combination with axitinib was tested. When administered alone, bromocriptine was found to not solely block tumor growth, but also to restore the balance between pro-angiogenic and anti-angiogenic factors, leading to the normalization of blood vessels [33]. This result is promising given the high tolerability and safety of dopamine receptor type 2 agonists and the fact that they are already used in human pituitary tumors due to their effects on tumor cells. Therefore, it would be interesting to test whether the effect of these agonists on blood vessel normalization is present in other pituitary tumor models and whether the same dosage and administration pattern are needed to obtain efficient vascular normalization. While the use of axitinib was found to improve vascular remodeling and restrained tumor growth in the same hemorrhagic prolactinoma model, only the combined use of bromocriptine and axitinib led to the suppression of intratumoral hemorrhage and restored blood vessel perfusion [33]. These observations are consistent with other studies that have shown complementary effects of the VEGF/VEGFR pathway inhibitors when combined with drugs targeting alternative pathways implicated in angiogenesis, and further highlight the importance of combination therapies when targeting angiogenesis [33,34,35].

## 3. Immune Infiltrative Cells

### 3.1. The Implications of Immune Populations in Pituitary Tumors

Within the TME, the roles of infiltrating immune cells and the molecules they secrete are complex, cell- and context-dependent, and can ultimately lead to either pro-tumorigenic or anti-tumorigenic effects [7]. While cytotoxic CD8+CD45RO+ memory T cells, CD4+ T helper 1 T cells, innate cytotoxic lymphocytes, natural killer cells, natural killer T cells, and M1-type tumor-associated macrophages (TAMs) are generally considered to be anti-tumorigenic, the myeloid-derived suppressive cells, M2-type TAMs, mast cells, CD4+CD25+Foxp3+ regulatory T cells, CD4+ T helper 2 T cells, and T helper 17 cells are usually considered to be pro-tumorigenic [6,7,36]. It is important to note that these pro- and anti-tumoral roles are not firmly defined and may vary in different tumor stages and types. As such, the immune TME needs to be carefully characterized in the different groups of pituitary tumors prior to understanding its exact role. To date, studies aimed at defining the immune TME of pituitary tumors are rare and mostly rely on the immunohistological characterization of cells based on a single or a few immune markers that are not sufficient to define the complexity of immune cell lineages (Table 2) [37,38,39,40,41,42,43,44]. Moreover, the classification of pituitary tumors (including the use of IHC for diagnosis) and of their aggressiveness varies across different studies, as does the interpretation of the immune infiltrate, with some studies considering the immune infiltrate in terms of either its presence or absence, while other studies have looked at the quantity or the intensity of the various markers. Therefore, the published studies do not yet permit a precise characterization of the immune landscape of pituitary tumors. For example, in a histological study on 35 human pituitary tumors (9 densely granulated growth hormone (GH) tumors, 9 sparsely granulated GH, 9 null cell, and 8 corticotroph tumors), Lu et al. examined the different immune cell populations. Throughout their study, they used CD68 staining to define macrophages and found these to be detected to various degrees in all of the cases. The number of CD68+ cells was positively correlated with tumor size and Knosp grade, and the authors found CD68+ cells to be more numerous in sparsely granulated GH and null cell tumors compared to densely granulated GH and corticotroph tumors [40]. While these observations are interesting, the fact that the sparsely-granulated GH and null cell tumors analyzed were larger than the densely-granulated GH and corticotroph tumors, and that the tumor size itself was positively correlated with the Knosp grade, may impact on their interpretation. Interestingly, these authors also looked at CD4+ cells (interpreted as CD4+ T cells) and CD8+ cells (interpreted as CD8+ cytotoxic T cells), with both cell types being more numerous in GH tumors than in null cell and corticotroph tumors [40].

The importance of TAMs in pituitary tumors has also been more recently highlighted in two more complex studies [43,45]. Based on a flow cytometry analysis of CD11b-expressing cells in 16 non-functioning pituitary tumors, it was found that tumors with greater than 10% CD11b+ cells on flow cytometry were the most expansile, having either a dimension >3.5 cm or a Ki67 staining index >3% [45]. These authors further found that tumors invading the cavernous sinus had an M2-type TAMs/M1-type TAMs ratio >1, while 80% of the non-invasive tumors had an M2-type TAMs/M1-type TAMs ratio <1. Moreover, using the THP-1 human monocyte cell line, they found that conditioned medium from M2-differentiated THP-1 cells promoted increased proliferation and migration of primary tumor cell cultures when compared to conditioned medium obtained from M1-differentiated THP-1 cells [45], suggesting that the presence of different types of TAMs in the TME may have distinct effects on the proliferation and invasiveness of pituitary tumors. Interestingly, culture media derived from primary cultures of three tumors also had different effects on macrophages, with two of them showing greater monocytes recruitment and polarization to an M2-type TAM phenotype, while the medium from one tumor caused less monocyte recruitment and resulted in polarization to an M1-type TAM phenotype [45]. These observations emphasize the bidirectional interactions that may exist between tumor cells and the TME. In parallel to this work, it was reported that among a group of somatotroph tumors, aryl hydrocarbon receptor-interacting protein (AIP)-mutated tumors showed an increased infiltration of CD68+ macrophages when compared to sporadic GH tumors [43]. Interestingly, using the rat GH3 somatomammotroph cell-line with AIP knockdown, they found that conditioned culture media collected from rat macrophage cultures induced a more prominent epithelial-to-mesenchymal transition-like phenotype, and enhanced migratory and invasive capacity in GH3 AIP-knockdown cells compared to control GH3 cells that retain AIP expression [43]. These findings are consistent with the reported role of TAMs in the epithelial-to-mesenchymal transition and in the invasive behavior of many tumor cell types [46]. More interestingly, the paper points out that the crosstalk between TAMs and AIP-mutated tumor cells is complex and may rely on an intricate interaction based on the C-C chemokine ligand 5 (CCL5)/C-C chemokine receptor type 5 (CCR5) pathway that appears to be important for the promotion of the invasive character of AIP-mutated GH tumors [43]. Although this is promising, further work is needed to determine whether any of the several emerging TAM-targeting strategies could be applied to pituitary tumor patients.

### 3.2. Immunotherapy in Pituitary Tumors

In recent years, many publications and clinical trials have confirmed the major importance of immunotherapies, especially immune checkpoint inhibitors, in cancer therapeutics. While anti-programmed cell death protein 1 (anti-PD-1) and anti-cytotoxic T-lymphocyte-associated protein 4 (anti-CTLA4) antibodies have been extensively used to target immune checkpoints in many cancers, their use in pituitary tumors has only just commenced, with a case of a corticotroph carcinoma bearing alkylator-induced hypermutations showing a partial response on combined nivolumab and ipilimumab [25] (Table 1). Based on our current knowledge, pituitary tumors have been shown to be infiltrated by T lymphocytes [40,41,42] and to express programmed death ligand 1 (PD-L1), which is considered a potential predictor of the response to immune checkpoint inhibitors [41,42]. Although these findings raise the hope of the potential efficacy of immune checkpoint inhibitors in pituitary tumors, it is too early to draw any conclusion, especially in view of the fact that such therapeutic strategies, though used in many cancer types, to date have proven effective in only a maximum of 20% of cases for most tumor types [8].

Currently, two clinical trials are running based on an ipilimumab and nivolumab combined therapy, one dedicated to pituitary tumors including carcinomas (NCT04042753), and one basket trial accepting only pituitary carcinomas (NCT02834013). As the only case treated so far was previously treated with alkylating agents, which by inducing somatic hypermutations may render the tumor responsive to immunotherapy [25], it will be particularly interesting to see the comparative effect of immune checkpoint inhibitors in patients who were previously treated (or not) with such therapies. Moreover, studies have shown that the effect of immune checkpoint inhibitors could be enhanced by the administration of concurrent radiotherapy [47,48]. Therefore, combining immunotherapy with radiotherapy may also become a therapeutic option in the treatment of pituitary tumors.

One should be careful when considering targeting the components of the TME, as such a network likely has complex multiple crosstalk between the TME and tumor cells, but also between different components of the TME itself. One such interaction is the promotion by hypoxia and abnormal blood vessels of a pro-tumoral immune TME, which in turn will promote tumor angiogenesis, leading to a vicious circle [8,30,49]. Angiogenesis has also been shown to be important in immunotherapy, as it may lead to resistance to immune checkpoint inhibitors [49]. Therefore, the combined use of immunotherapies and drugs targeting angiogenesis is emerging as a novel strategy for the treatment of numerous cancers [8,49]. Interestingly, such observations may also have important implications for the treatment of pituitary tumors as it was recently reported that pituitary tumors invading the cavernous sinus show a higher expression of VEGF, VEGFR1, and PD-L1, as well as having a higher number of TAMs [44]. This suggests that, at least in the case of the aggressive non-functioning pituitary tumors analyzed in their study, tumor angiogenesis is associated with an immunosuppressive TME, and that immunotherapy combined with therapies aimed at obtaining vascular normalization should be envisioned in such tumor types.

## 4. Resident Folliculostellate Cells

Folliculostellate (FS) cells are resident non-endocrine cells that comprise 5–10% of the normal anterior pituitary and are also found in the TME of pituitary tumors [50,51,52,53,54]. They are identified based on morphological criteria (either having star-like processes and/or forming follicles when they come together) and on their immunoreactivity for the S100 protein [55,56]. Since the identification of the S100 protein as a marker, FS cells have been shown to be immunoreactive to other markers such as glial fibrillary acidic protein (GFAP), major histocompatibility (MHC) class II surface antigens, cytokeratins, and vimentin, and to produce numerous bioactive molecules [56,57,58,59]. Due to the differential expression of these markers and molecules, FS cells are currently considered to be phenotypically and functionally heterogeneous [54,56,57], with three main subtypes being proposed: astrocyte-like (also expressing GFAP), dendritic cell-like (also expressing MHC class II), and epithelial cell-like (also expressing cytokeratins) [57,59,60,61]. In the normal anterior pituitary, FS cells are mainly associated with the regulation of the hormone secretion of endocrine cells [57], but they have also been implicated in the microcirculation of ions, nutrients, and waste products [58]. Moreover, FS cells have been shown to mediate and modulate the neuroendocrine response to inflammation and immune stress [56,62], and to have phagocytic properties [63,64]. FS cells are further associated with the production of numerous cytokines and growth factors, including interleukin 6 (IL6), follistatin, basic fibroblast growth factor, transforming growth factor β, VEGF, leukaemia inhibitory factor [58], and macrophage inhibitory factor [62]. In addition, both adherent and gap junctions are found between FS cells and between FS cells and endocrine cells [56,65]. These observations support the strong implication of FS cells in the intercellular communication and paracrine exchanges within the anterior pituitary.

Besides the roles FS cells have in the normal anterior pituitary, their identification in pituitary tumors suggests FS cells may also have major implications in these tumors. However, to date, the available knowledge concerning FS cells in human pituitary tumors comes mainly from IHC studies assuming their presence via S100 immunoreactivity alone, while their role in the initiation, maintenance, and progression of pituitary tumors or in the functioning phenotype of these tumors remains largely undetermined.

In one of the biggest studies to date, Voit et al. studied FS cells in 286 cases of pituitary tumors in patients presenting with acromegaly. The authors reported that FS cells were found to be either isolated or grouped, forming network-like structures and were frequently in close relationship with the tumor cells [52]. Their results showed that 198 out of 286 tumors (69%) contained FS cells, with the number and distribution of FS cells varying between tumors: 35% of the tumors (100 cases) presented few widely sparse FS cells, 15% of the tumors (43 cases) presented FS cells scattered throughout the tumor, and 19% of the tumors (55 cases) presented abundant FS cells. When they examined clinical correlations, these authors found the existence of a negative correlation between the density of FS cells and the preoperative mean prolactin levels [52], an observation rather contradictory to published in vitro work that generally showed that the interaction between prolactin-secreting cells and FS cells, and/or their secreted molecules, results in an increased prolactin level [57,66]. The same authors also found that preoperative mean growth hormone levels were higher in patients with tumors containing few widely sparse (64.5 ± 8.1 μg/L) or scattered FS cells (83.1 ± 17.1 μg/L) than in those patients with tumors lacking FS cells (44.9 ± 4.1 μg/L) [52]. This observation supports the possibility that part of the role of FS cells in pituitary tumors may be similar to the function of FS cells in the normal pituitary, where one of their roles is to regulate the secretory capacity of normal endocrine pituitary cells. However, it is interesting that the difference observed with the widely sparse or scattered FS cells was not observed in the case of abundant FS cells. Indeed, the preoperative mean growth hormone level was lower in patients with tumors containing abundant FS cells (41.0 ± 5.7 μg/L) than in patients with tumors without FS cells (44.9 ± 4.1 μg/L) [52]. This may suggest that during TME remodeling, the function of these cells is modified or that different types of FS cells are recruited.

In another study, Vajtai et al. started by analyzing three cases of pituitary tumors, two prolactinomas, and one gonadotroph tumor, which presented with an inflammatory reaction mediated by T lymphocytes that selectively involved the tumoral tissue. These authors first observed that perivascular T lymphocytes (predominantly CD4+) tended to mingle with cells immunopositive for the S100 protein. In order to test whether the FS cells from the inflammatory foci presented monocytic/dendritic properties, they performed double immunohistochemical staining for both the S100 protein and the MHC class II antigen HLA-DR and they showed that in these three cases, many of the FS cells co-expressed both epitopes. These cells that could not be morphologically distinguished from the FS cells negative for the MHC class II antigen HLA-DR were distributed both between tumoral acini and alongside intratumoral vessels, mingling with the T lymphocytes. CD1a (as a marker for Langerhans’ cells) and CD21 (as a marker for follicular dendritic cells of lymphoid type), as well as cytokeratins, tested negative. Moreover, no FS cell co-expressing S100 protein and HLA-DR was observed in the peritumoral tissue in these 3 cases, nor in the tumoral tissue of another 48 cases of pituitary tumors lacking an inflammatory reaction. The authors postulated that an appropriate inflammatory TME may induce an FS cell subset to adopt a dendritic cell-like phenotype and that these cells may have an antigen presentation function in pituitary tumors [50].

To understand the mutual interactions between the tumor and the peritumoral tissue during tumor progression, Farnoud et al. [51] looked at the boundary between the tumor and the adjacent normal anterior pituitary tissue in a series of 18 pituitary tumors. These authors reported the presence of a transition zone with a modified architecture between the tumor and the peritumoral tissue. Interestingly, they found the density of the FS cells to be higher in the transition zone and its close vicinity compared to the density observed in the tumor center or in the normal pituitary tissue distant from the tumor. In addition, alterations of the basement membrane were observed in the peritumoral tissue adjacent to this transition zone. These observations, coupled with older observations, notably the increased activity shown by FS cells under pathological conditions, their phagocytic activity, and their capacity to secrete angiogenic growth factors, led the authors to suggest that FS cells may be involved in basement membrane remodeling, tumoral neo-angiogenesis, and tumoral expansion [51].

In 2000, the first FS cell line derived from a human gonadotroph pituitary tumor was established and named PDFS, for pituitary-derived FS cells [67]. PDFS were demonstrated to show an epithelial-like morphology and to express S100 protein and vimentin [67]. This cell line has since been used mainly to study autocrine and paracrine communications in the human pituitary. PDFS cells express prolactin-releasing peptide, produce bioactive activin A and follistatin, and have an intact activin signaling pathway [66,67]. Using PDFS, it was also shown that glucocorticoids upregulate annexin-1 synthesis that is afterwards translocated to the cell surface and exert paracrine regulatory functions on adrenocorticotropic hormone (ACTH) release by corticotroph cells, thus being implicated in the regulation of the hypothalamo–pituitary–adrenal axis [68].

The presence of FS cells in the TME of pituitary tumors and their association with clinical traits support their potential role in tumorigenesis-related processes, but a better understanding of their heterogeneity and their functions in tumors is still needed before we can assess their potential use as a therapeutic target. Interestingly, although in other organs (such as the pancreas), resident stellate cells are also present and these stellate cells have been shown to have the potential to become cancer-associated fibroblasts (CAFs) [69,70,71], which are considered to be a major component of TME in numerous cancers [6,72], the possibility of FS cells or a subset of FS cells becoming pituitary tumor-associated fibroblasts has not yet been explored. This possibility would be particularly appealing given the abundant existing literature on CAFs and on ways to target them, a topic that has been recently reviewed [72].

## 5. Cancer/Tumor-Associated Fibroblasts

CAFs have been shown to be phenotypically and functionally heterogeneous. Both resident cells (like stellate cells, resting fibroblasts, and endothelial cells) and infiltrative cells (like bone marrow-derived precursors) can become CAFs [73,74]. From a functional perspective, CAFs are currently seen as a key component of the TME, and while they have been implicated mostly in tumor progression, they can also exhibit cancer-restraining functions [74,75]. CAFs can produce and secrete growth factors, cytokines, chemokines, extracellular matrix (ECM)-remodeling enzymes, and ECM components. To that extent, they are implicated in a wide range of processes that include tumor growth, cancer stemness, angiogenesis, ECM remodeling, metabolic and immune reprogramming of the TME, tumor invasion and metastasis, and even in chemoresistance [6,73,74,75]. All these processes are of potential interest to pituitary tumors and carcinomas, and the study of pituitary tumor-associated fibroblasts will hopefully provide valuable insight into the biology and treatment of these tumors in the future. Unfortunately, to date, the literature on the subject is extremely scarce and limited to a single paper [76] and one conference abstract [77].

Using co-cultures and mouse subcutaneous xenografts of GH3 somatomammotroph cells and fibroblasts derived either from invasive pituitary tumors, non-invasive pituitary tumors, or from the sphenoid sinus mucosa of patients bearing non-invasive pituitary tumors (fibroblasts considered as normal by the authors), Lv et al. found that the CAF-marker α-smooth muscle actin (α-SMA), as well as VEGF, showed higher expressions in tumor-associated fibroblasts derived from invasive pituitary tumors than in the ones derived from non-invasive pituitary tumors or normal fibroblasts. Furthermore, in their experiments, tumor-associated fibroblasts derived from invasive pituitary tumors promoted the proliferation of GH3 cells in vitro and the growth of GH3-derived xenographs in mice. In addition, VEGF expression was higher in the grafted tumors when GH3 cells were co-injected with tumor-associated fibroblasts derived from invasive pituitary tumors than when co-injected with tumor-associated fibroblasts derived from non-invasive pituitary tumors or with normal fibroblasts [76]. The fact that tumor-associated fibroblasts derived from invasive tumors promoted in vitro and in vivo tumor growth and showed VEGF overexpression is not surprising, but the study has several limitations nonetheless, one being the lack of positive and negative controls for the immunostaining for α-SMA.

Other than this paper, another team has also recently communicated on pituitary tumor-associated fibroblasts. Although with the obvious limitation of having only an abstract available, their results are encouraging. This unpublished work confirms the possibility of isolating tumor-associated fibroblasts from both somatotroph and non-functioning pituitary tumors, and also the IHC detection of tumor-associated fibroblasts in pituitary tumors. Moreover, by studying the secretome of tumor-associated fibroblasts derived from pituitary tumors, the authors found, for example, that the fibroblasts derived from invasive pituitary tumors were secreting higher levels of IL6 compared to the fibroblasts derived from non-invasive pituitary tumors. In addition, when they treated these fibroblasts with pasireotide (based on their somatostatin receptor expression), the secreted IL6 decreased by 80%. Furthermore, they mentioned that GH3 somatomammotroph cells showed a morphology epithelial-to-mesenchymal-like and increased invasiveness when treated with conditioned media derived from tumor-associated fibroblasts than when treated with control-conditioned media produced by normal skin fibroblasts [77]. Yet, these two examples highlight the limited work that has been done so far to comprehend the role of pituitary tumor-associated fibroblasts, pinpointing the need to pursue the study of their exact implication in pituitary tumors.

From a therapeutic point of view, this study suggests that somatostatin analogues currently in use for pituitary tumors may act on both tumor cells and pituitary tumor-associated fibroblasts [77]. This is consistent with some observations made regarding pancreatic cancer, where the use of pasireotide has been shown to inhibit protein synthesis (including IL6 production) in αSMA positive CAFs. Moreover, when combined with gemcitabine chemotherapy, pasireotide reduced the CAF-secretome-triggered chemoresistance of cancer cells and the tumor growth [78]. From the wider literature, many more potential therapeutic strategies targeting CAFs are emerging for cancer treatment. These range from CAF depletion via cell surface marker targeting, to altering CAF activation or functions (as pasireotide does) and to CAF normalization (for example by pharmacologically stimulating the vitamin D receptor) [72].

## 6. Composition and Remodeling of the Extracellular Matrix in Pituitary Tumors

The ECM is a three-dimensional network of macromolecules (proteins, glycosaminoglycan, proteoglycans, and glycoproteins) in which soluble molecules, such as growth factors and chemokines, are embedded [6,79,80,81]. The ECM is highly dynamic and is fundamental to both physiological and pathological processes, being implicated in the adhesion, migration, proliferation, growth, polarity, differentiation, survival, and apoptosis of cells [81,82,83]. In the anterior pituitary, the ECM has been shown to play key roles in both the normal gland and in pituitary tumors [80,83], while both endocrine and non-endocrine cells of the anterior pituitary have been shown to regulate the ECM composition, which appears to be different in the normal anterior pituitary compared to ECM in pituitary tumors [80,83,84,85].

So far, most studies on pituitary tumors have examined the expression of matrix metalloproteinases (MMPs), enzymes that are implicated in the degradation and remodeling of the ECM and in releasing the molecules embedded in it [6], and at tissue inhibitors of metalloproteinases (TIMPs), which regulate MMPs. In a study that included 54 pituitary tumors [86], it was reported that invasive pituitary tumors expressed increased levels of MMP-2 and MMP-9 mRNA and protein compared to non-invasive tumors [86]. Through the analyses of 82 pituitary tumors, Hui et al. showed that the expression of *MMP-1*, *-2*, *-9*, *-14*, and *-15* genes were elevated in invasive pituitary tumors compared to non-invasive pituitary tumors or normal anterior pituitary glands, highlighting *MMP-14* as the gene with the highest expression. They then performed IHC only for MMP-2 and -14 and confirmed increased levels of these two proteins in invasive pituitary tumors [7]. In prolactinomas, MMP-9 expression, assessed using IHC, was shown to be more likely to be present and at a higher density in invasive versus non-invasive tumors. Prolactinomas positive for MMP-9 also showed higher vascular densities. Interestingly, the same authors have also examined the expression of MMP-9 in non-recurring versus recurring non-functioning tumors at their first and second presentation. Although no difference was found regarding the presence of MMP-9 at the first presentation between tumors that recurred later or those that did not, the same tumors were more likely to show the expression of MMP-9 at their second presentation [87]. Other authors reported that in addition to the higher mRNA and protein expression levels of MMP-9 found in invasive pituitary tumors, these tumors also showed a decreased expression of TIMP-1 [88]. More importantly, their work highlights that increased serum levels of MMP-9 and decreased serum levels of TIMP-1 can be quantified in patients bearing invasive pituitary tumors [88]. Similar observations have been made by other authors, who detected a decreased level of TIMP-1 in conditioned medium derived from pituitary tumors when compared to that from normal pituitary explants [89]. Analysis of the expression of the four members of the TIMP family (*TIMP-1*, *-2*, *-3*, and *-4*) performed in normal anterior pituitaries and pituitary tumors showed that *TIMP-3* was the most abundant TIMP in normal anterior pituitaries and that *TIMP-3* expression positively correlated with the fibrous matrix deposition found in pituitary tumors [85].

From a functional perspective, it is interesting to note that apart from their role in ECM remodeling, MMPs may also have actions that impact tumor cells. This hypothesis is indeed supported by Paez-Pereda et al., who showed in in vitro experiments using pituitary tumor cell lines (AtT-20 and GH3), that the inhibition of MMPs resulted in the inhibition of hormone secretion and of cell proliferation only when growth factors were embedded in the matrigel, demonstrating that MMPs acted by releasing these growth factors [89]. Based on the above observations, the remodeling of the ECM by MMPs therefore appears likely to play an important role within the TME of pituitary tumors.

Besides the MMP system, Knappe et al. also investigated the expression of the serine proteases urokinase-type plasminogen activator (uPA), the tissue-type plasminogen activator (tPA), the uPA receptor (uPAR), and the plasminogen activator inhibitor-1 (PAI-1) in a large series of 84 pituitary adenomas (18 cases of acromegaly, 21 Cushing’s diseases, 18 prolactinomas, 1 TSH-secreting adenoma, and 26 non-secreting adenomas) and 9 normal pituitaries. These authors found a diffuse expression of uPA (89% of the cases), tPA (69% of the cases), uPAR (90% of the cases), and PAI-1 (87% of the cases) in pituitary adenomas. ACTH-secreting tumors showed an overexpression of uPAR when compared to all other pituitary tumors. In the normal anterior pituitary, tPA, uPAR, and PAI-1 were focally expressed in all samples, while uPA was weakly expressed in six out of nine samples and absent in three. In ACTH-secreting tumors, there was an overexpression of tPA in non-invasive tumors as compared to the invasive ones. These authors found that pituitary tumors showed a higher uPA expression compared to normal anterior pituitary tissue. They further found that invasive non-secreting tumors had a tendency toward the overexpression of uPA compared to their non-invasive counterparts, suggesting that the uPA system may have a role in the dural invasion of pituitary tumors [90]. These observations are consistent with studies from other cancer types where the uPA system is a key player in the breakdown of ECM components, and in which the elevated expression of uPA and of uPAR is observed and associated with poor prognosis [91].

Therefore, both the MMP system and the uPA system seem to be important for pituitary tumor biology, and especially for invasiveness. Saeger et al. have even proposed in 2016 that at least the detection of MMP-2, MMP-9, TIMP-2, and of uPA seemed justified [92]. However, one should be careful when asserting invasiveness, as the extension of pituitary tumors into the cavernous sinus may result both from tumor expansion and/or tumor invasion [92,93]. Moreover, these enzymes might theoretically show a different distribution inside the same tumor (for example in the center versus the infiltrative areas), but careful consideration must be given to such observations because during transsphenoidal pituitary surgery, the sellar tumor areas are easily accessible for sampling, whereas the areas of infiltration into the cavernous sinus are rather sucked away, resulting in possible sampling artefacts [92].

Besides MMPs and serine proteases, other components of the ECM have also been shown to play various roles in pituitary tumors. First, regarding the distribution of the main ECM macromolecules, there are differences between normal and tumoral pituitary tissue. Farnoud et al. showed that in the normal anterior pituitary, fibronectin, laminin, and collagen IV were present in the basement membranes (both the parenchymatous one limiting the endocrine cell cords and the endothelial one) and in the connective tissue [94,95], while collagen I was present only in the connective tissue. In PAs, on the other hand, the authors found that the parenchymatous basement membrane was either fragmented or completely absent and associated with immunoreactivity to fibronectin and collagen I, which were also found in the wall of the vessels [94]. Another ECM molecule, vitronectin, was found in the TME of pituitary tumors but was not present in the connective tissue of normal anterior pituitaries [95]. Moreover, other authors analyzed the IHC expression of collagenous and non-collagenous ECM macromolecules in the perisellar connective tissue of 10 normal pituitary glands as these macromolecules may also serve as potential targets for tumor-derived proteinases and be important for the cavernous sinus infiltration. In the boundaries of the sella and around the cavernous sinus, the authors found strong expression of collagen I, collagen III, and fibronectin; moderate expression of collagen IV, tenascin, and vitronectin; and weak expression of collagen II, while the expression of laminin and elastin was weak or absent. Interestingly, the authors also noticed that collagen IV, fibronectin, vitronectin, and tenascin showed condensation in the pituitary capsule [96].

From a functional perspective, the fact that laminin expression was shown to progressively decrease from the normal anterior pituitary stage to lactotroph hyperplasia and again to the prolactinoma stage in mice and in humans, combined with its capacity to inhibit prolactin production and GH3 cell proliferation, supports its role in the actual tumorigenesis of prolactinomas [97]. In the corticotroph AtT-20 cell line, laminin, as well as fibronectin and collagen I have been shown to inhibit ACTH secretion. Moreover, in AtT-20 cells, laminin and collagen I were also shown to inhibit cell proliferation, while fibronectin and collagen IV exerted the opposite effect [98]. Interestingly, in both experiments, the same ECM components did not affect the secretion of prolactin and ACTH from normal rat pituitary cells, potentially reflecting a change in the expression of corresponding receptors (i.e., integrins) during tumorigenesis [80,97,98]. Indeed, in a study on 26 pituitary tumors and 6 normal pituitaries, a number of integrins (the membrane receptors that transduce signals from the ECM to the cell) were shown to be differently expressed in pituitary tumors compared to normal tissue; specifically, all hormone-producing cells and FS cells from the normal anterior pituitary expressed integrins α_3_β_1_ and α_6_β_4_, while in tumor cells, α_3_β_1_ expression was downregulated, α_6_β_4_ expression abrogated, and neoexpression of α_ν_β_3_ was observed in three of the tumors. Moreover, a greater range of integrin subunits was observed in the stroma of tumors (α_1_, α_3_, α_5_, α_6_, α_ν_, β_1_, β_3_, β_4_, β_5_) compared to the connective tissue found in the normal pituitary (α_1_, α_5_, β_1_) [95], and the expression of the integrin-β1 subunit correlated with pituitary tumor invasiveness [99]. Finally, the exposure of GH3 cells to different collagen subtypes has been shown to differently affect their invasion ability and migration strategy in vitro, highlighting the functional complexity that ECM composition may have in pituitary tumors [100].

The role of ECM molecules in controlling the biophysical properties of the stroma (such as molecular density, topography, rigidity/stiffness, and tension) may also prove relevant for pituitary tumors since it has already been shown that characteristics such as tissue stiffness (which usually increases in malignant tissues) play a major role in other cancers [79,101,102,103]. In the pituitary field, our knowledge of pituitary tumor stiffness is rather limited and has mainly been assessed from a surgical point of view (i.e., the preoperative predictive value of different imaging techniques), as firm fibrous tumors, representing <20% of the resected pituitary tumors, are more difficult to resect through a transsphenoidal approach than the soft tumors, which represent the majority of pituitary tumors [104,105,106,107,108]. The assessment of tumor collagen content, as evaluated by Masson’s staining, has shown that the consistency of the tumor as evaluated by the surgeon was associated with the collagen content, with firm tumors having more collagen [105,106,107]. While statistical analysis associating tumor consistency and the invasion of the cavernous sinus did not show significant differences between soft and firm tumors in a series of 34 pituitary tumors [107], it would appear important to assess clinico-pathological associations in larger series and also to more precisely map the actual stiffness landscape of those tumors through atomic force microscopy (AFM), as has already been done for normal anterior pituitary tissue [109].

Lastly, while work is still to be done to fully understand both the composition and contribution of the ECM within pituitary tumors, potential treatment targets, such as MMPs, are already emerging as appealing candidates.

## 7. Concluding Remarks

The understanding of the TME in pituitary tumors is still in its infancy. Nevertheless, taking into account the existing data, as well as the much greater knowledge we have on other tumor types, there is no doubt that the TME represents a major component of pituitary tumors that holds much promise for targeted therapies. Therefore, better and more systematic characterization of the composition and role of the TME in all pituitary tumor subtypes (i.e., based on their hormone type/cell lineage, their functional status, and their genetic profile) is urgently needed. It is of major importance to consider the TME as an intricate and collaborative network of molecules and cells that could influence tumor behavior and ultimately influence the response to treatments. We believe it is important to keep in mind that in addition to the different TME molecules, cells, and associated mechanisms we have presented here, the hypothalamic hormones, as well as pituitary hormones produced by tumor cells and normal pituitary endocrine cells that are still present in pituitary tumors or in their close vicinity, may also play important endocrine, paracrine, and/or autocrine roles in remodeling the TME. Examples of this are the reported role of prolactin in the stimulation of angiogenesis [110] and of GH in the epithelial-to-mesenchymal transition seen in various tumor types [111], while antagonists of GH-releasing hormone receptors themselves have been shown to inhibit tumor growth and progression in different types of tumors [112,113,114].

Prior to developing and testing TME therapeutics, it should also be kept in mind that the TME is a dynamic structure that evolves with the tumor and may present intra- and inter-patient heterogeneity. Therefore, dedicated personalized strategies may need to be developed to target tumors at different stages in their progression in individual patients. To that end, defining the most appropriate and efficient strategies will require a precise map of TME in pituitary tumors to be drawn. This will not be a trivial task as one of the biggest challenges to arise will be the understanding of the inter-patient and intra-tumor TME heterogeneity, but also incorporating the recent emerging concept of the tumor macroenvironment. This concept underlines the interplay existing between the tumor and the whole organism, including the age of the patient and the systemic metabolic and inflammatory status, amongst other factors [115]. The tumor macroenvironment may also prove to be highly relevant for pituitary tumors in the future, especially given that abnormalities in pituitary hormone levels may influence systemic metabolism [116], as well as systemic immune function [117].

In conclusion, while recent progress in our understanding of the TME of pituitary tumors has been made and this has started to pave the way for the development of innovative therapeutics, this review highlights the need to further improve our knowledge of the complex micro- and now macro-environments of pituitary tumors in order to provide effective and personalized treatment for pituitary tumor patients in the future.

## Figures and Tables

**Table 1 cancers-11-01605-t001:** Cases of aggressive pituitary tumors and pituitary carcinomas treated with TME-targeting therapies.

Sex and Age at Diagnosis	Tumor Type	Previous Tumor-Directed Treatments	TME-Targeting Therapy	Response to the TME-Targeting Therapy	Ref.
Male, 38 years	Silent corticotroph carcinoma	4 NS, RT, NS, TMZ, NS, TMZ, surgery for metastases, TMZ, RT for metastases, NS	Bevacizumab 10 mg/kg every 2 weeks for 26 months (ongoing)	Tumor volume: stable disease for 26 months	[19]
Female, 25 years	Functioning corticotroph carcinoma	3 NS, RT, somatostatin analogue, RT, TMZ	Bevacizumab + pasireotide for 6 months	Tumor volume: stable disease at 6 months Hormonal secretion: plasma ACTH ↘ from >200,000 pmol/L to 113,000 pg/mL at 6 months	[20]
Female, 50 years	Functioning corticotroph adenoma	2 NS, RT, 3 NS, lanreotide + cabergoline, TMZ	Bevacizumab*	Transient stable disease (patient deceased due to complications of another NS)	[21]
Male, 63 years	Functioning corticotroph carcinoma	NS	Bevacizumab 10 mg/kg every 2 weeks + TMZ 75 mg/m^2^ daily + RT for 2 months (followed by 12 cycles of TMZ)	Tumor volume: complete response of the pulmonary nodule 8 weeks after the start of bevacizumab + TMZ + RT (no recurrence for 5 years)	[22]
NA	NA	No previous TMZ*	Bevacizumab + first-line TMZ *	Partial response	[23]
NA	NA	TMZ*	Bevacizumab + second course of TMZ *	Partial response	[23]
NA	NA	TMZ*	Bevacizumab + second course of TMZ *	Progressive disease	[23]
NA	NA	TMZ*	Bevacizumab + second course of TMZ *	NA	[23]
NA	NA	TMZ*	Bevacizumab as a second-line therapy *	Partial response after 3 months	[3,23]
NA	NA	TMZ*	Bevacizumab as a second-line therapy *	Stable disease	[3,23]
NA	NA	TMZ*	Bevacizumab as a third-line therapy *	Progressive disease	[3,23]
Male, 50 years	Functioning corticotroph adenoma transformed into silent corticotroph carcinoma	NS, RT, surgery for metastases, RT for metastases, TMZ	Bevacizumab 10–15 mg/kg every 2 weeks (09/2010–11/2012) + TMZ 150–200 mg/m^2^ daily for 5 consecutive days monthly (09/2010–08/2011)	Tumor volume: stable disease for 8 years	[24]
NA	NA	TMZ*	Sunitib *	Progressive disease	[23]
Female, 35 years	Functioning corticotroph carcinoma	2 NS, RT, 2 NS, pasireotide, cabergoline, TMZ + capecitabine (4 cycles), TMZ + capecitabine (2 cycles), etoposide + carboplatin (2 cycles), RT for the primary tumor	Nivolumab 1 mg/kg + ipilimumab 3 mg/kg every 3 weeks (5 cycles), followed by maintenance nivolumab	Tumor volume: ↘59% (primary tumor) and ↘92% (main liver metastasis) after the 5 cyclesHormonal secretion: plasma ACTH ↘ from 45,550 to 66 pg/mL after the 5 cycles (59 pg/mL at the 6-month follow-up)	[25]

Tumor microenvironment (TME), reference (Ref.), neurosurgery (NS), radiotherapy (RT), temozolomide (TMZ), adrenocorticotropic hormone (ACTH), not available (NA), * no further information available.

**Table 2 cancers-11-01605-t002:** Immunohistochemical studies of the immune microenvironment of human pituitary tumors.

Marker and Cell Type	Number and Type of Tumors	Study Type	Quantification	Prevalence	Associations/Correlations with Anatomoclinical Characteristics	Ref.
Y182A (* macrophages)	27	IHC (frozen sections)	Semi-quantitative scale (various fields 40× objective): 1 (occasional positive cells), 2 (up to 20), 3 (20–40), and 4 (>40)	Tumor proper 100% (mean rating 1.2), perivascular 92% (mean rating 1.5)	NA	[37]
RFTCT: CD2, CD3, CD7 + CD8Y (* T cells)	23 and 21			Tumor proper 47% (mean rating 0.4), perivascular 80% (mean rating 0.8)		
CD8 (* T cells)	24			Tumor proper 33% (mean rating 0.3), perivascular 66% (mean rating 0.7)		
CD4 (* T cells)	28: 6 GH, 3 GH-PRL, 2 GH-ACTH, 2 FSH, 2 PRL, 1 LH, 1 ACTH, 2 NIR, 9 NA			Tumor proper 7% (mean rating 0.07), perivascular 14% (mean rating 0.1)		
RFBCT: CD20 + RFBT (* B cells)	23			Perivascular 1/23 (occasional positive cells)		
Leu-11b (* NK)	13			1/13 (PRL)		
LCA/CD45 (* lymphocytes)	1400: 411 PRL, 137 GH, 166 ACTH, 15 TSH, 42 FSH-LH, 44 αSU, 275 NIR, and 310 multihormonal	IHC (paraffin-embedded sections)	First, lymphocytic infiltrate diagnosed histologically at 400× magnification if ≥15–20 lymphocytes present by counting the nuclei on whole sections, then IHC	40/1400 (2.9%)	Presence associated with lower age (37 vs. 41) in all tumors, but not when PRL only considered; presence: PRL > NIR, and αSU > NIR, ACTH and multihormonal	[38]
CD45R0 (* T cells)				NA, but lymphocytes were almost exclusively CD45R0+	NA	
CD20 (*B cells)				NA, but very rare		
CD45 (* lymphocytes)	72: 40 secreting (14 ACTH, 18 GH, 4 PRL, 4 TSH), 32 non-secreting + 12 NPG (autopsy) + 14 autoimmune hypophysitis	IHC (FFPE sections)	Semi-quantitative scale: 0 (only few positive cells) to 5 (most of the tissue infiltrated by positive cells)	18/72 (25%), score 0.5 or 1 (13/18) and score 2 or 3 in 5/18 (2 PRL and 3 non-secreting)	Presence associated with poor clinical outcome; prognostic factor for tumor persistence/recurrence independent of tumor size	[39]
CD68 (* macrophages)	35: 9 densely granulated GH, 9 sparsely granulated GH, 9 null cell, and 8 ACTH	IHC (FFPE sections)	Average positive cell number on 10–30 consecutive fields (original magnification 400×)	35/35 (100%), varying degrees	Number positively correlated with tumor size and Knosp classification; more numerous in sparsely granulated GH and null cell tumors than in densely granulated GH and ACTH tumors	[40]
CD4 (* T cells)				Rare to sparse	More numerous in GH tumors than in null cell and ACTH tumors	
CD8 (*T cells)						
CD20 (* B cells)	densely granulated GH, sparsely granulated GH, null cell, and ACTH		NA	Essentially absent	NA	
CD45	48: 28 functioning (PRL and GH) and 20 non-functioning (silent gonadotroph and null cell)	IHC (tissue microarrays)	Mean intensity of the positive IHC staining (intensity/area)	NA, variable expression	IHC staining intensity: tumors with proliferative indices >3% > tumors with proliferative indices ≤3%	[41]
CD3 (* T cells)					IHC staining intensity: functioning > non-functioning tumors	
CD4 (* T cells)						
CD8 (* T cells)					None	
PD-1					IHC staining intensity: non-functioning > functioning tumors, and tumors with proliferative indices >3% > tumors with proliferative indices ≤3%	
PD-L1		IHC (tissue microarrays) + RNAscope *in situ* hybridization	IHC: mean intensity of the positive staining (intensity/area); RNAscope: average number of dots per cell	48/48 (100%), variable expression	Transcript level and IHC staining intensity: functioning > non- functioning tumors; transcript level: primary tumors > recurrent tumors	
CD8 (* T cells)	191: 106 non-functioning, 40 PRL, 31 GH, 9 ACTH, and 5 plurihormonal	IHC (FFPE sections)	Positivity = cytoplasm or membrane staining in >5% of tumor cells	166/191 (86.9%)	Positivity associated with PRL tumors (not with functioning tumors when considered together), and with higher blood levels of GH	[42]
PD-L1				70/191 (36.6%)	Positivity associated with functioning tumors when considered together, with PRL and GH tumors when subtypes considered separately, with higher blood levels of PRL, GH, ACTH, and cortisol, with a Ki67 index ≥3.0%, and with the CD8+ staining	
CD68 (* macrophages)	26: 9 AIP-mutated GH, 17 sporadic GH, and 9 NPG (autopsy)	IHC (FFPE sections and tissue microarrays)	% of cells (3–5 random fields at 400× magnification)	NA	More numerous in AIP-mutated GH tumors than in sporadic ones and NPG	[43]
FOXP3 (* regulatory T cells)	26: 9 AIP-mutated GH, 17 sporadic GH, and 11 NPG (autopsy)					
CD8 (* T cells)	29: 12 AIP-mutated GH, 17 sporadic GH, and 11 NPG (autopsy)			NA (some positivity in tumors, while negative staining in NPG)	None	
CD45RO (*T cells)						
CD163 (* M2-type macrophages)	27 non-functioning: 17 with carvernous sinus invasion and 10 without	IHC (FFPE sections)	3 selected hot spots on low-power fields (4×), then positive cells counted in these areas using high-power fields (40×)	NA	More numerous in invasive tumors than in non-invasive tumors for the carvernous sinus	[44]
FOXP3 (* regulatory T cells)				NA	Foxp3/CD8+ cells ratio higher in invasive tumors (and with tendency of more numerous CD8+ cells) than in non-invasive tumors for the carvernous sinus	
CD8 (* lymphocytes)				NA		
CD4 (* lymphocytes)				NA	None	
PD-1			NA	NA	None	
PD-L1			Expression in: ≥50% of tumor cells (score 3+); <50% but ≥5% of tumor cells (2+); <5% but ≥1% of tumor cells (1+); <1% of tumor cells (0)	NA	The score tended to be higher (*p* = 0.050) in the carvernous sinus invasion group: score 2 or 3 in 8 patients, score 0 or 1 in 9 patients, while in the group without carvernous sinus invasion: score 3 in one patient, score 0 or 1 in 9 patients	

Natural killer cells (NK), reference (Ref.), growth hormone (GH), prolactin (PRL), adrenocorticotropic hormone (ACTH), follicle-stimulating hormone (FSH), luteinizing hormone (LH), non-immunoreactive (NIR), not available (NA), thyroid-stimulating hormone (TSH), α-subunit (αSU), normal pituitary glands (NPG), aryl hydrocarbon receptor-interacting protein (AIP), immunohistochemistry (IHC), formalin-fixed paraffin-embedded (FFPE), * interpreted as.

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
