# Peer review of "The Microenvironment of Pituitary Tumors—Biological and Therapeutic Implications"

_cancers, 2019, doi:10.3390/cancers11101605_

Round 1
Reviewer 1 Report
The manuscript presents a very comprehensive review of the literature regarding the role of TME in PitNETs pathogenesis and targeted therapy and can help for better understanding the PitNETs in order to further studies in this direction for providing an effective and personalized treatment for the patients. The manuscript is well written. I suggest improving the manuscript with more graphical representations.
For a more coherent way of presentation of the manuscript, I suggest the division the subchapter 2: angiogenesis in the tumor microenvironment in 2 subchapters: 1. The implication of angiogenesis in PitNETs tumorigenesis and to add also a schematic representation of the findings and 2. Therapeutic targeting of angiogenesis in PitNETs
The same scheme can be applied also for the chapter immune infiltrative cells, resident folliculostellate cells and composition and remodeling of the extracellular matrix in PitNETs.
I suggest to add also a chapter regarding cancer-associated fibroblasts in PitNETs, even if several studies reviewed their implication in pathogenesis and targeted therapy of several cancers, but you can summarize somehow the findings relevant for PitNETs.
My recommendation is to accept with minor revision.
Reviewer 2 Report
The authors present an interesting and timely overview of the current knowledge on the tumor microenvironment (TME) of pituitary neuroendocrine tumors (PitNETs). There are some points to be addressed that would make the review still more valuable.
“ … the association between angiogenesis and clinical traits such as invasiveness …” (lines 79-80): this should be more elaborated (general aspects), particularly because PitNETs are often invasive. Please briefly describe the Stupp protocol for the non-expert. It has been shown that the FS-cell population also contains immune-type cells (such as dendritic and macrophage-like cells; see papers by Allaerts et al.). Can the authors also relate this character to a possible immune role in PitNETs? What is still missing is an overview of the TME’s cytokines/growth factors that may play a role in pituitary tumors. Moreover, FS-cells are well-known producers of many of these molecules. Please give a brief overview of the literature. Mentioning “Author et al.” is not always needed in the text. Prevalence of carcinomas should be specified in the introduction. Lines 89-97: the impact of bevacizumab sounds overstated (only 1 patient showing complete remission).Author Response
Please see the attachment.

Reviewer 3 Report
This review presents an interesting and well written overview on different aspects of the tumor microenvironment (TME) of pituitary adenomas and carcinomas which the authors term pituitary neuroendocrine tumors (PitNETs). Whether this namingmakes sence in th econtext of thi sspecial manuscript seems to be douptful, since the authors quote different targeted antitumorous therapies, but never refer ton NETs, respectively. The tables give a useful summary informatiob on the few patients with aggressive pituitary adenomas or carcinomas who underwent targeted therapies. The authors admit that "understanding of TME in PitNETs is still in its infancy", but give a very helpful overview on the current kowledge of angiogenesis, immune infiltartive cells, resident folliculostellate cells, extracellular matrix (ECM), and possible interactions between them in pituitary adenomas. E.g. the concept to achieve vascular normalization and the possible value of combination therapies are inspiring for further research in this field.
Concerning ECM and the data on proteinases the authors focus on metalloproteinases only and may consider also the possible role of different expression of serin proteinases in different subtypes of pituitary adenomas (1). Moreover, the authors should comment on the different distribution of collagenous and non-collagenous ECM components in the stroma and the covering of the pituitary gland (2). They claim, that there may be different interactions between the TME components in the center the infiltrative periphery of pitutary tumors. One problem in this kind of research are possible sampling errors, since during transsphenoidal surgery tissue samples from the center of the tumors are easily obtained, whereas in the area of infiltration to the cavernous sinus for safety reasons the tumors most often is sucked away. Parasellar infiltration of pituitary tumors, however, is common and represents the main reason for incomplete tumor removal (3). Until now it remains unclear, why this intravenous infiltration is only raraely associated with the development of hematogeneous metastases, and why pituitary adenomas and carcinomas do not differ histologically (4). Therefore, this review is a valuable contribution to the lieterture on tumorigenesis and infiltrative growth of pituitay adenomas and carcinomas. It may help researchers who are interested in this field to get an overview on current knowledge and interactions of TME components.
Knappe UJ et al. Acta Neurochir 2010, 152:345-53 Micko AS et al. J Neurosurg 2015, 122:803-11 Saeger W et al. Endocr Pathol 2016, 27:115-22
